# Enhanced Deep-Learning-Based Automatic Left-Femur Segmentation Scheme with Attribute Augmentation

**DOI:** 10.3390/s23125720

**Published:** 2023-06-19

**Authors:** Kamonchat Apivanichkul, Pattarapong Phasukkit, Pittaya Dankulchai, Wiwatchai Sittiwong, Tanun Jitwatcharakomol

**Affiliations:** 1School of Engineering, King Mongkut’s Institute of Technology Ladkrabang, Bangkok 10520, Thailand; 63601005@kmitl.ac.th; 2King Mongkut Chaokhunthahan Hospital (KMCH), King Mongkut’s Institute of Technology Ladkrabang, Bangkok 10520, Thailand; 3Division of Radiation Oncology, Department of Radiology, Faculty of Medicine Siriraj Hospital, Mahidol University, Bangkok 10700, Thailand; pittaya.dan@mahidol.ac.th (P.D.); wiwatchai.sit@mahidol.ac.th (W.S.); tanun.jit@mahidol.ac.th (T.J.)

**Keywords:** deep learning, automatic segmentation, femur bone, U-Net, attribute augmentation, cropping

## Abstract

This research proposes augmenting cropped computed tomography (CT) slices with data attributes to enhance the performance of a deep-learning-based automatic left-femur segmentation scheme. The data attribute is the lying position for the left-femur model. In the study, the deep-learning-based automatic left-femur segmentation scheme was trained, validated, and tested using eight categories of CT input datasets for the left femur (F-I–F-VIII). The segmentation performance was assessed by Dice similarity coefficient (DSC) and intersection over union (IoU); and the similarity between the predicted 3D reconstruction images and ground-truth images was determined by spectral angle mapper (SAM) and structural similarity index measure (SSIM). The left-femur segmentation model achieved the highest DSC (88.25%) and IoU (80.85%) under category F-IV (using cropped and augmented CT input datasets with large feature coefficients), with an SAM and SSIM of 0.117–0.215 and 0.701–0.732. The novelty of this research lies in the use of attribute augmentation in medical image preprocessing to enhance the performance of the deep-learning-based automatic left-femur segmentation scheme.

## 1. Introduction

### 1.1. Background

Medical image segmentation involves segmenting or annotating regions of interest in a medical image such as conventional X-ray, computed tomography (CT) scans, magnetic resonance imaging (MRI), and ultrasound images. In medical image segmentation, a medical image is divided into distinct regions that correspond to anatomical structures of interest, with the goal to divide anatomical structures within the image into separate regions for medical applications, particularly in cancer radiotherapy planning [1,2].

In radiotherapy planning, image segmentation (or contour delineation) is required to distinguish between tumors and organs at risk (OARs), which are healthy organs or tissues that may be adversely affected by the radiation treatment. This is achieved in order to determine the optimal radiation dose and direction of the beam in the treatment to shrink tumors or eradicate cancer cells while sparing the nearby healthy tissue [3,4,5,6]. As a result, contour delineation is vital for effective cancer treatment [7,8]. Evidence shows that 48.3% of all cancer cases are treated by radiotherapy [9], suggesting that image segmentation is of vital importance in the treatment of cancers.

Contour delineation or segmentation was traditionally carried out by manually tracing the boundaries before the development of algorithm-driven semi-automatic technology for medical image segmentation. However, the disadvantage of the manual and semi-automatic medical image segmentation methods is the overreliance on physicians’ experience, making the results subject to intra- or inter-observer variability [10,11,12].

Manual and semi-automatic contour delineation is time-consuming and labor-intensive because one scan dataset (i.e., one medical image) consists of hundreds of slices. The contouring time varies depending on factors such as tumor volume, the number of OARs, the complexity of regions of interest, and the beam angle [7]. The contouring time lasts between 30 min–12 h depending on cancers and stages, e.g., 5–10 h for manual contouring of abnormal lesions using the neuroimaging scans [13], 1.5–3 h for head and neck radiotherapy [14], and 3 h for the intensity-modulated radiation therapy treatment planning [15]. The lengthy contouring or segmentation time results in a backlog of cancer cases, making it difficult to provide timely treatment to all patients who need it.

To address the issue of lengthy contouring time, a fully automatic medical image segmentation scheme based on deep learning technology has been developed. Specifically, deep-learning-based automatic medical image segmentation improves efficiency (i.e., shortened radiotherapy planning time) and reliability (i.e., improved segmentation accuracy) while reducing the workload of physicians [16].

Deep learning plays a significant role in the Fourth Industrial Revolution (i.e., Industry 4.0), specifically in healthcare [17]. Of particular relevance is convolutional neural network (CNN), which is a class of artificial neural network that is commonly utilized in medical image segmentation due to the algorithm’s applicability to various image quality, shapes, and sizes. Moreover, CNN involves blocks of series of operation layers (e.g., convolutional, pooling, and transposed convolution layers), resulting in excellent pattern recognition [18]. CNN automatically extracts relevant features from the training dataset for a required task by iteratively adjusting its weights with backpropagation. As a result, the CNN-based outcomes are superior to the manual outcomes [19]. In addition to CNN-based methods, transformer-based methods have also emerged as prominent approaches in medical image segmentation. These methods leverage the attention mechanism to selectively weigh the importance of different parts of the input data. In [20], it was identified that the standard measurement of the distance from the tumor’s lowest boundary to the anal verge is insufficient. As a result, a novel method was proposed to automatically measure the distance to anal verge (DTAV), accompanied by the design of a boundary-guided transformer for accurate rectum and tumor segmentation. In [21], recognizing the transformer model’s ability to capture extensive global information and its reliance on pre-training on large-scale datasets, a hybrid CNN-transformer network (HCTNet) was proposed, consisting of transformer encoder blocks (TEBlocks) in the encoder and a spatial-wise cross Attention (SCA) module in the decoder. Furthermore, ref. [22] proposed a hierarchical hybrid vision transformer (H2Former) by integrating the merits of CNNs, multi-scale channel attention, and transformers for medical segmentation.

### 1.2. Related Works

Despite the high reliability of the manual segmentation of medical images, the method is subject to the intra- and inter-observer variability. In response to this issue, semi-automatic segmentation techniques have been developed, which integrate mathematical algorithms such as thresholding, level set [23], seeded region growing, localized region-based active contour model, clustering-based methods, K-means, and clustering-based methods [24]. While these techniques have improved efficiency and reliability [25,26], they still suffer from inter-observer variability and are unsuited for certain medical images.

In order to address these limitations, deep learning models have been integrated into automatic medical image segmentation technology. Preprocessing plays a critical role in enhancing the performance of deep-learning-based automatic segmentation by transforming data into a format that is more readily processed [27]. The preprocessing methods proposed to enhance the performance of deep-learning-based automatic segmentation include window leveling, filtering, matching, histogram techniques [28], T1, FLAIR (skull stripping) [29], wavelet decomposition, local binary patterns [30], region of interest (ROI) selection, bias field correction, resampling methods [31], normalization [28,29,30,31], and crop ROI [32,33].

Evidence shows that the segmentation accuracy is significantly enhanced through a combination of preprocessing techniques [28,29,31,34]. Therefore, this study utilizes both conventional and novel preprocessing techniques to enhance image contrast and detect target organs, including window leveling, histogram projection, cropping, and attribute augmentation.

However, the conventional preprocessing techniques are less useful in certain cases, such as detecting or locating the target organs when the Hounsfield units (HU) of the organs of interest are similar or identical to those of the nearby organs and/or when the target organs are paired organs, e.g., femurs, kidneys, and lungs. As a result, attribute augmentation can be used to enhance the performance of the deep-learning-based automatic left-femur segmentation scheme. There are many previous works that have used data augmentation techniques and attribute-aware methods to improve object recognition and segmentation performance in various domains [35,36,37,38,39,40]. In [35], the authors aimed to improve the performance of object recognition and segmentation tasks by introducing a novel attribute-aware feature encoding (AFE) module within a multi-task network while enhancing semantic attributes through the integrated approach of attribute-aware feature encoding and the regularization of feature encoding via auxiliary attribute learning. To overcome the limitations posed by occlusions, varying lighting conditions, and objects with similar visual appearances, ref. [36] proposed a method that combines attribute-aware techniques and data augmentation to boost the performance of semantic segmentation methods. To address the scarcity of annotated data in the medical domain and overcome the associated challenges of high cost and effort in manual annotation, [37] introduced the cycle-consistent cross-domain medical image segmentation (CyCMIS) method, which leverages cycle-consistent techniques and diverse image augmentation to enhance the transferability, robustness, and generalization of segmentation models, enabling more accurate and reliable segmentation results even with limited labeled data in the target domain.

Addressing the inherent challenges of capturing significant variations in size, shape, texture, and color of skin lesions, ref. [38] introduced a method that combines multi-scale convolutional neural networks (CNNs) and domain-specific augmentations, involving specific transformations and enhancements applied to skin lesion images to simulate realistic variations, with the goal of enhancing the segmentation of skin lesions and their attributes. Aimed to enhance the generalization capability of deep learning models, improving segmentation performance, and addressing the limited availability of annotated training data, the authors of [39] developed an innovative approach that combines K-means clustering, deep learning techniques, and synthetic data augmentation. This approach involves generating synthetic data to augment the limited annotated data and improve the segmentation performance of deep learning models. To develop and evaluate an algorithm for bone segmentation on whole-body CT using a convolutional neural network (CNN), [40] proposed a method that utilizes a CNN along with novel data augmentation techniques, including conventional methods, mixup, and random image cropping and patching (RICAP).

In contrast, this research proposes an innovative data augmentation method that combines attribute-aware techniques, multiple regression theory [41,42,43], and deep learning models for medical image segmentation. The proposed method enhances the diversity and realism of synthetic images by utilizing a domain-specific, knowledge-driven data augmentation strategy.

Of particular interest is the U-Net model, which achieves high segmentation performance and is used in both architectural [33,34] and non-architectural aspects [13,28,29,30,31] of medical image segmentation.

The research methodology of this study follows [28,29,30,31] with minor modifications. However, unlike [28,29,30,31], this research relies on the lower abdominal CT scans of Thai patients, with the permission of the Siriraj Institutional Review Board.

Specifically, a notable correlation was observed between lying posture and the position of the left–right femur, indicating that the patient’s lying position can influence the femur’s positioning. The lying posture, including supine and prone postures, is utilized as data attributes in the experiments. This research proposes augmenting cropped CT slices with data attributes during image preprocessing to improve the performance of the deep-learning-based automatic left-femur segmentation scheme. In the study, the deep-learning-based automatic left-femur segmentation scheme was trained, validated, and tested using eight categories of CT input datasets (F-I–F-VIII).

The segmentation performance of the deep-learning-based automatic left-femur segmentation scheme was determined by Dice similarity coefficient (DSC) and intersection over union (IoU). The similarity between the predicted 3D reconstruction images of the left femur and ground-truth images were measured by spectral angle mapper (SAM) and structural similarity index measure (SSIM). The novelty of this research lies in the use of attribute augmentation in medical image preprocessing to enhance the performance of the deep-learning-based automatic left-femur segmentation scheme.

The organization of this research paper is as follows: Section 1 is the introduction. Section 2 describes the U-Net segmentation model for the left femur. Section 3 details the experimental dataset and data preprocessing. Section 4 deals with the segmentation performance of the proposed deep-learning-based automatic left-femur segmentation scheme and the image similarity metrics, and Section 5 compares and discusses the experimental results. The conclusions are provided in Section 6.

## 2. Deep-Learning-Based Automatic Left-Femur Segmentation Scheme: U-Net Segmentation Model

The U-Net segmentation model for the left femur comprises a five-layer, fully connected convolutional neural network (Figure 1). The U-Net is a type of CNN architecture consisting of a series of distinct operation layers, e.g., convolutional and pooling layers. The operation layers transform the input volume (e.g., the input image or another feature map) into an output volume (e.g., mask images and the feature maps) through a differentiable function. Figure 1 illustrates the U-Net architecture, which contains two paths: the contraction path (left side) and the expansion path (right side).

As shown in Figure 1, the CT input datasets are first entered into the contraction path and convolved and maximum-pooled (i.e., undergoing blocks of series of operation layers). Specifically, one block of series of operation layers consists of two consecutive convolutional layers and one max-pooling layer.

In each operation layer, the input datasets or feature maps are first padded to expand the size to allow for the center element of the kernel (3 × 3 kernel) to be placed over every pixel in the source, with a stride of 1, and each source pixel is then replaced with a weighted sum of the respective source pixel ($x_{i}$) and nearby pixels, where the weight ($w_{i}$) of the convolution kernel is a learnable parameter. The bias (b) is then added to the weighted sum of the source pixel and nearby pixels ($\sum_{i=0}^{j}w_{i}x_{i}$) before applying the activation function ($f_{A}$) to obtain *Z*, as expressed in Equation (1). The activation function ($f_{A}$) between convolutional layers is the rectified linear unit (ReLU), as expressed in Equation (2).
(1)$$Z=f_{A}(\sum_{i=0}^{j}w_{i}x_{i}+b)$$
where j is the size of kernel.
(2)$$f_{ReLu}\left(x\right)=\left\{\begin{array}{c} 0\;\mathrm{for}\;x\;<\;0\\ x\;\mathrm{for}\;x\;\geq \;0 \end{array}\right.$$

The aforementioned convolution process is repeated for a new feature map before maximum-pooling, given 2 × 2 filter and a stride of 1, to downsample and avoid overfitting. In the contraction path, the feature map resolution of each block of series of operation layers is halved, while the channels are doubled.

Meanwhile, the expansion path of the U-Net architecture restores spatial information for high-resolution feature maps and extracts features. In the expansion path, one block of series of operation layers consists of one transposed convolution, one concatenation, and two consecutive convolutional layers. The transposed convolution layer (3 × 3 convolution kernel and a stride of 2) doubles the feature map resolution and halves the channels. Moreover, the feature map (output) of the transposed convolution layer is concatenated with the corresponding feature map from the contraction path to compensate for missing features (i.e., skip connection). The process is repeated until the resolution of the feature map is identical to that of the CT input dataset.

In the final operation layer (represented by the yellow arrowhead), the feature map of the last convolutional layer is convolved, given 1 × 1 kernel convolution, to reduce the feature channels (64) to one channel or class output (i.e., the left femur). In addition, the sigmoid activation function (Equation (3)) is used to distribute the probability of output pixels and then binarily classified as 0 and 1, where 0 denotes non-target organs (depicted in black) and 1 the target organs (i.e., the left femur; depicted in white).
(3)$$f_{sigmoid}\left(x\right)=\frac{1}{1+e^{-x}}$$

## 3. Experimental Dataset and Data Preprocessing

Table 1 tabulates the initial experimental datasets, which are of CT-slice datasets in 8-bit PNG format from 120 CT scans of the lower abdomen of 120 patients (60 male and 60 female patients) of 60–80 years of age and with lower abdominal diseases including colorectal cancer, cervical cancer, prostate cancer, rectosigmoid cancer, and rectum cancer.

The CT-slice datasets are used to train, validate, and test the deep-learning-based automatic left-femur segmentation scheme. In radiotherapy of the lower abdomen, the OARs of radiation exposure are the bladder, left–right femur, prostate, rectum, and small bowel. Specifically, the target OARs of the deep-learning-based segmentation scheme is the left femur.

In this research, a CT scan consists of a series of CT slices (cross-sectional images) in the axial (horizontal), coronal (frontal), and sagittal (longitudinal) planes. The 3D CT slices were acquired using SOMATOM Confidence^®^ 32-slice CT simulator (Siemens, Germany) in the HELIX operation (helical scanning mode) with 120 kV, 250 mA, and 3 mm slice thickness. The use of the CT scans was reviewed and approved by the Siriraj Institutional Review Board with Certificate of Approval (COA) no. Si 315/2021.

Prior to preprocessing, the 120 CT scans were annotated to create an extensive dataset of source image–ground-truth mask image pairs and verified by radiologists. As shown in Figure 2, the data preprocessing entails four major steps: (i) assigning xyz coordinates of the bounding box for cropping, and this step involves window leveling and histogram projection; (ii) contrast enhancement using window leveling and cropping the CT slices; (iii) augmenting the cropped CT slices with attributes; and (iv) dividing the preprocessed datasets into three groupings: datasets for training, validation, and testing the deep-learning-based left-femur segmentation scheme. In this study, the purpose of cropping is to resize the image while retaining the image quality, whereas the aim of attribute augmentation is to enhance the predicted result of the deep-learning-based left-femur segmentation scheme.

### 3.1. Assigning xyz Coordinates of the Bounding Box for Cropping

In the femur cropping, the grayscale of each CT slice of the 120 CT scans was adjusted by using window leveling and normalization to distinguish the target OARs from the surrounding tissues and organs [44,45]. Window leveling is a viewer setting affecting the range of grayscale or Hounsfield unit (HU) numbers in the image. Window leveling and normalization is carried out by adjusting the window level (*WL*), which is the midpoint of the range of the HU numbers and the window width (*WW*), which is the measure of the range of minimum and maximum HU (${HU}_{min}$ and ${HU}_{max}$). ${HU}_{min}$ and ${HU}_{max}$ can be calculated by Equations (4) and (5), respectively [46].
(4)$${HU}_{min}=\left(WL-0.5\right)-(\frac{WW-1}{2})$$
(5)$${HU}_{max}=\left(WL-0.5\right)+(\frac{WW-1}{2})$$

In this research, *WL* and *WW* are 300 HU and 400 HU, which are the optimal *WL* and *WW* for maintaining the entire HU of the bone [47,48] (Figure 3). Given *WW* of 400 HU, an HU less than or equal to ${HU}_{min}$ (i.e., ≤100 HU) is designated as $y_{min}$ and HU greater than or equal to ${HU}_{max}$ (i.e., ≥499 HU) as $y_{max}$. In this study, $y_{min}$ and $y_{max}$ are 0 HU and 255 HU, respectively. The HU between ${HU}_{min}$ and ${HU}_{max}$ ($y_{i})$, which is in the range of 0 to 255 HU, is calculated by Equation (6) [46].
(6)$$y_{i}=(\left(\frac{x_{i}-\left(WL-0.5\right)}{\left(WW-1\right)}\right)+0.5)\times \left(y_{max}-y_{min}\right)+y_{min}$$
where $y_{i}$ is the HU output in the range of 0 to 255 HU (i.e., 2^8-bit^), and $x_{i}$ is the HU input given that i denotes the number of pixels (i.e., the resolution of image) of the CT slice. Since the preprocessed CT slice is in 8-bit PNG format, the range of the HU output ($y_{i}$) is between 0–255 HU.

The final task in assigning the xyz coordinates of the bounding box for femur cropping involves identifying the xyz coordinates by using histogram projection to locate the target organ (i.e., the femur). There exist several deep-learning-based preprocessing techniques to detect or locate organs of interest, e.g., bone fracture detection [49]. In the histogram projection, the vertical histogram projection is calculated by summing all rows of HU (${HU}_{r}$) of a given column of the CT slice (Equation (7)) and the horizontal histogram projection by summing all columns of HU (${HU}_{c}$) along a given row of the CT slice (Equation (8)) [50,51,52].
(7)$$Vertical\;Histogram\;Projection\;=\frac{\sum {HU}_{r}}{Sc}$$
(8)$$Horizontal\;Histogram\;Projection\;=\frac{\sum {HU}_{c}}{Sc}$$
where c is the number of columns, r is the number of rows, and Sc is a scaling value to normalize the output of vertical histogram projection and horizontal histogram projection. In this research, Sc is set at 100.

Given the minimum thresholds for the vertical and horizontal histogram projections of 50 HU in the axial plane and 5 HU in the coronal and sagittal planes, the potential coordinates of all CT slices in all planes were determined (i.e., xy, xz, and zy coordinates for the axial, coronal, and sagittal planes). The smallest xy, xz, and zy coordinates (among the potential coordinates) of each CT slice were selected, and the axis coordinates of identical plane were averaged for the xyz coordinates of the CT scans. We obtained 120 xyz coordinates, corresponding to 120 CT scans of patients with lower abdominal cancers.

Figure 4 shows, as an example, the horizontal and vertical histogram projection and the xy coordinates of the femur in the axial plane. Figure 5 shows the workflow for assigning the xyz coordinates of the bounding box for femur cropping.

### 3.2. Contrast Enhancement and Femur Cropping

To enhance the contrast, CT slices are window-leveled given WL and WW of 120 HU and 336 HU, which are the optimal WL and WW for CT image contrast [53], respectively, and normalized into the range of 0 HU to 255 HU.

Figure 6a,b show the original CT slice in the axial plane and the corresponding CT image in 8-bit PNG format after window leveling and normalization. As seen in Figure 6b, the window leveling and normalization noticeably improve the image quality.

The xyz coordinates (in Section 3.1) are applied onto the contrast-enhanced CT slices to delineate the bounding box for femur cropping, as shown in Figure 7a. The purpose of cropping is to resize the image while retaining the image quality. The size of the cropped CT slices of the femur is 360 × 200 pixels in the axial plane, 360 × 110 pixels in the coronal plane, and 200 × 110 pixel in the sagittal plane. Figure 7b depicts, as an example, the bounding box for cropping (represented by the red rectangle) and the cropped cube comprising the cropped CT slices in the axial, coronal, and sagittal planes.

### 3.3. Cropped CT Slices Augmented with Attributes (Feature Addition)

Attribute augmentation (or feature addition), based on multivariate regression in machine learning [41,42,43], is utilized to enhance the segmentation performance of the deep learning algorithm. In this research, the attribute augmented to the cropped CT slices is lying position (supine and prone posture), which was selected based on expert suggestion from one of the co-authors, who is also an oncologist. Lying position was included as a data attribute because the position of the patient during CT scanning can affect the shape and position of internal organs and target volumes, such as the femur and rectum (e.g., [54,55]). Accounting for these differences can potentially improve segmentation accuracy, and by including lying position as a data attribute, the model can capture these variations. Attribute data are expected to affect the segmentation performance of the deep learning U-Net model.

Figure 8 shows the cropped CT slice of the femur before and after attribute augmentation.

In order to differentiate between the CT slices of different lying positions (when patients entered the CT scanner), the feature coefficients of the lying position attribute are categorized into three cases: small (1 and 2 for supine and prone posture, respectively), large (5 and 10), and excessively large (10 and 20). Since the prone position is unique to rectum cancer disease, where the lesion is located at the back, it is treated as a special case that influences the position and characteristics of the left–right femur, as shown in Figure 9. This can potentially confuse the segmentation model. Therefore, the coefficient value of the prone posture was assigned to be higher than the coefficient value of the supine posture, allowing for a significant differentiation between a special case and a normal case. Essentially, the feature coefficients were varied to investigate the effect of coefficient values on the performance of the U-Net segmentation model.

Figure 10 shows, as an example, the augmentation of the attribute to the cropped CT slices, where the white and yellow matrices represent the cropped CT slice and the attribute. The attribute augmentation is performed by adding M × 1 matrix to the last column of the cropped CT slice, where M is the number of rows. The attribute-augmented CT slices are subsequently converted into CT input datasets in 8-bit PNG format.

### 3.4. Training, Validation, and Testing Datasets of the Deep-Learning-Based Left-Femur Segmentation Scheme

The initial CT scans of the lower abdomen (before preprocessing) belong to 120 patients with lower abdominal cancers. The proportion of input datasets for training, validation, and testing the deep-learning-based automatic left-femur segmentation scheme is 60:20:20. Specifically, out of 120 CT scans, 72 CT scans were used for training, 24 CT scans for validation, and 24 CT scans for testing.

In addition, the deep-learning-based segmentation scheme was trained, validated, and tested under eight categories of CT input datasets, namely F-I–F-VIII for the left-femur segmentation.

Category F-I refers to the uncropped and non-augmented CT-image input datasets of the left femur; F-II to the cropped CT-image input datasets of the left femur (without attribute augmentation); and F-III, F-IV, and F-V to the cropped and augmented CT-image input datasets of the left femur with small (1 and 2 for supine and prone posture), large (5 and 10), and excessively large feature coefficients (10 and 20), respectively; and F-VI, F-VII, and F-VIII to the uncropped and augmented CT-image input datasets of the left femur with small (1 and 2 for supine and prone posture), large (5 and 10), and excessively large feature coefficients (10 and 20), respectively.

In the training, the weight ($w_{i}$) and bias (b) of the U-net model for left-femur segmentation was optimized by gradient descent optimization given the binary cross-entropy loss function and the learning rate (α) and epoch of 0.001 and 5000. The iteration was terminated when IoU [56] fails to improve for 50 consecutive epochs. Table 2 tabulates the hyperparameters of the U-Net models for left-femur segmentation.

The experiments were carried out using the APEX system of CMKL University, Thailand. APEX is a high-performance computing platform and storage infrastructure for AI works, with 1920 vCPU cores, 48x A100 GPUs with 1.92 TB total GPU memory, 30 petaFlops AI, 7.5 TB System Memory, 3 petaBytes Storage, and 200 Gbs Interconnect.

## 4. Segmentation Performance and Image Similarity Metrics

The segmentation performance of the deep-learning-based automatic left-femur segmentation scheme was assessed by DSC (Equation (9)) and IoU (Equation (10)). Specifically, DSC focuses on the prediction performance (i.e., segmentation accuracy) on average of the deep learning model, whereas IoU focuses on the worst prediction performance of the algorithmic model [57].
(9)$$DSC\left(A,B\right)=\frac{2\left|A\cap B\right|}{\left|A\right|+\left|B\right|}$$
(10)$$IoU\left(A,B\right)=\frac{\left|A\cap B\right|}{\left|A\cup B\right|}=\frac{\left|A\cap B\right|}{\left|A\right|+\left|B\right|-\left|A\cap B\right|}$$
where A is the ground-truth CT image region, and B is the predicted region.

In addition, 3D reconstruction images were rendered based on the predicted results from the deep-learning-based segmentation models for a visually distinct comparison. The similarity between the 3D reconstruction images and ground-truth images were evaluated by SAM (Equation (11)) and SSIM (Equation (12)). Both metrics are image-similarity measures that quantify the degree of visual and semantic similarity of a pair of images.
(11)$$\mathrm{SAM}\;(\alpha )={cos}^{-1}\frac{\sum_{i=1}^{nb}t_{i}r_{i}}{\sqrt{\sum_{i=1}^{nb}{t_{i}}^{2}}\sqrt{\sum_{i=1}^{nb}{r_{i}}^{2}}}$$
where *t* is the predicted 3D reconstruction image pixel spectrum, and *r* is the 3D ground-truth image pixel spectrum in an *n*-dimensional feature space, *nb* is the number of bands in the 3D image, and α is the angle between the two spectra of the predicted 3D model reconstruction image and 3D ground-truth image. A small α indicates similarity between the predicted 3D reconstruction image and the 3D ground-truth image [58].

The SSIM is a perceptual metric that quantifies image-quality degradation caused by processing such as data compression or by losses in data transmission. The SSIM algorithm extracts three key features from an image: luminance (μ), contrast (σ), and structure; and the comparison between the two images (i.e., the predicted 3D reconstruction image and 3D ground-truth image) is performed on the basis of these three features. The SSIM is mathematically expressed in Equation (12) [59,60].
(12)$$SSIM\left(x,y\right)=\frac{\left(2\mu_{x}\mu_{y}+c_{1})(2\sigma_{xy}+c_{2}\right)}{\left(\mu_{x}^{2}+{\mu_{y}}^{2}+c_{1})(\sigma_{x}^{2}+\sigma_{y}^{2}+c_{2}\right)}$$
where $\mu_{x}$ and $\mu_{y}$ are the luminance of the predicted 3D reconstruction image (x) and the 3D ground-truth image (y), $\sigma_{xy}$ is the correlation coefficient between the two images $,\sigma_{x}$ and $\sigma_{y}$ are the contrast of the predicted 3D reconstruction image (x) and the 3D ground-truth image (y), and $c_{1}$ and $c_{2}$ are constants to avoid an undefined value when ${\mu_{x}}^{2}+{\mu_{y}}^{2}$ or $\sigma_{x}^{2}+\sigma_{y}^{2}$ is approaching zero.

The SSIM is in the range of −1 to 1, where −1 indicates that both images (i.e., predicted 3D reconstruction image and 3D ground-truth image) are dissimilar, and 1 indicates that both images are identical.

## 5. Results and Discussion

This section discusses the segmentation performance of the proposed deep-learning-based automatic left-femur segmentation scheme. The testing of the deep-learning-based segmentation scheme was carried out under eight categories of the input datasets of CT images, namely F-I–F-VIII for the left-femur segmentation.

Specifically, category F-I refers to the uncropped and non-augmented CT-image input datasets of the left femur; F-II to the cropped CT-image input datasets of the left femur (without attribute augmentation); and F-III, F-IV, and F-V to the cropped and augmented CT-image input datasets of the left femur with small (1 and 2 for supine and prone posture), large (5 and 10), and excessively large feature coefficients (10 and 20), respectively; and F-VI, F-VII, and F-VIII to the uncropped and augmented CT-image input datasets of the left femur with small (1 and 2 for supine and prone posture), large (5 and 10), and excessively large feature coefficients (10 and 20), respectively.

### 5.1. Performance of the U-Net Femur Segmentation Model

Table 3 tabulates the DSC and IoU of the U-Net left-femur segmentation model under the eight categories of the CT-image input datasets: categories F-I–F-VIII.

Under dataset category F-I, the DSC and IoU of the left-femur segmentation model are 37.76% and 23.90% and 67.96% and 52.32% under category F-II. The DSC and IoU are 61.37% and 45.46%, 88.25% and 80.85%, and 72.54% and 57.93% under categories F-III, F-IV, and F-V, respectively. The DSC and IoU are 51.37% and 41.25%, 48.62% and 35.18%, and 45.27% and 31.30% under categories F-VI, F-VII, and F-VIII, respectively. The segmentation performance of the left-femur segmentation model decreases as the feature coefficients increase beyond a certain limit, as indicated by DSC and IoU under category F-V compared to category F-IV.

The DSC and IoU under categories F-VI, F-VII, and F-VIII are poorer than under categories F-II, F-III, F-IV, and F-V. The finding could be attributed to the substantially larger CT image size under categories F-VI, F-VII, and F-VIII (512 × 512 pixels) compared to under categories F-II, F-III, F-IV, and F-V (360 × 200 pixels). Specifically, the optimal feature coefficients for the U-Net left-femur segmentation model, given the dataset under category F-IV, are 5 for supine posture and 10 for prone posture, as evidenced by DSC and IoU of 88.25% and 80.85%.

Figure 11a–h, as an example, compare the performance of the U-Net left-femur segmentation model under dataset categories F-I, F-II, F-III, F-IV, F-V, F-VI, F-VII, and F-VIII, where the left, middle, and right columns are the CT image of the femur, the corresponding ground truth, and the predicted segmentation image. In Figure 11a (under dataset category F-I), the U-Net left-femur segmentation model displays the incomplete left femur compared to the ground-truth image. In Figure 11b (under dataset category F-II), the U-Net segmentation model displays the right femur (i.e., the non-target organ) in addition to the left femur (the target organ).

In Figure 11c (under dataset category F-III), the U-Net segmentation model displays small sections of the right femur, while certain sections of the left femur are missing. In Figure 11d (under dataset category F-IV), the U-Net segmentation model displays only the left femur (i.e., the target organ), and the predicted result closely resembles the ground-truth image. Meanwhile, in comparison with category F-IV, the predicted result under dataset category F-V is less complete, with certain sections of the left femur missing (Figure 11e). In Figure 11f, the U-Net segmentation model displays the incomplete left femur and certain sections of the right femur. In Figure 11g, the head of the left femur is missing, and in Figure 11h, the femur head of the target organ (left femur) is missing, and some sections of the right femur appear on the segmented image.

### 5.2. Comparison and Similarity of 3D Reconstruction Images

Table 4 compares the predicted 3D reconstruction images of the left femur and the ground-truth images and the corresponding SAM and SSIM under categories F-I, F-II, F-III, F-IV, F-V, F-VI, F-VII, and F-VIII.

The SAM and SSIM are 0.209–0.257 and 0.433–0.544 under category F-I; 0.135–0.236 and 0.658–0.714 under category F-II; 0.120–0.222 and 0.668–0.729 under category F-III; 0.117–0.215 and 0.701–0.732 under category F-IV; 0.121–0.223 and 0.675–0.729 under category F-V; 0.146–0.225 and 0.572–0.644 under category F-VI; 0.157–0.225 and 0.551–0.637 under category F-VII; and 0.184–0.236 and 0.461–0.639 under category F-VIII. By comparison, the predicted 3D reconstruction images using the cropped and augmented CT-image input datasets of the left femur with large feature coefficient values (category F-IV) closely resemble the ground-truth images, as evidenced by smallest SAM and largest SSIM.

Figure 12, Figure 13, Figure 14, Figure 15, Figure 16, Figure 17, Figure 18 and Figure 19 depict the 3D reconstruction images of the deep-learning-based automatic left-femur segmentation scheme under various categories of CT datasets of the left femur.

In Figure 12 (under dataset categories F-I), the left femur is almost entirely missing, with a small section of the non-target organ appearing in the 3D reconstruction image. In Figure 13 (under dataset categories F-II), a large section of the right femur (non-target organ) appears on the 3D reconstruction image, suggesting that the CT datasets require further preprocessing. In Figure 14 (under dataset categories F-III), there remain certain sections of the right femur, while the head and the distal of lesser trochanter of the left femur are missing.

In Figure 15 (under dataset categories F-IV), there remain small sections of the right femur (non-target organ), and the head of the left femur is slightly missing. In Figure 16 (under dataset categories F-V), there remains one tiny section of the right femur, but the head and the distal of lesser trochanter of the left femur are missing.

In Figure 17 (under dataset categories F-VI), the head of the left femur and the distal of lesser trochanter are missing, while large sections of the right femur (non-target organ) appear on the 3D reconstruction image. In Figure 18 (under dataset categories F-VII), larger sections of the head of the left femur and the distal of lesser trochanter are missing, and larger sections of the right femur appear on the 3D reconstruction image.

In Figure 19 (under dataset categories F-VIII), the distal of lesser trochanter is almost complete, while the femoral head of the left femur is missing, with some sections of the right femur appearing on the image. Essentially, the optimal CT datasets for the deep-learning-based automatic left-femur segmentation scheme are those belonging to category F-IV.

## 6. Conclusions

This research proposes augmenting cropped CT slices with data attributes during image preprocessing to enhance the performance of a deep-learning-based automatic left-femur segmentation scheme. The data attribute is the lying position (supine and prone posture) for the segmentation model. In the study, the deep-learning-based automatic left-femur segmentation scheme was trained, validated, and tested under eight categories of CT input datasets for the left femur (F-I–F-VIII). The segmentation performance of the left-femur segmentation scheme was evaluated by DSC and IoU, and the similarity between the predicted 3D reconstruction images and ground-truth images was measured by SAM and SSIM. The results show that the left-femur segmentation model achieved the highest DSC (88.25%) and IoU (80.85%) under category F-IV (using the cropped and augmented CT input datasets with large feature coefficients). Moreover, the SAM and SSIM of the left femur segmentation model are 0.117–0.215 and 0.701–0.732 under category F-IV. The optimal CT dataset for the deep-learning-based automatic left-femur segmentation scheme is that of dataset category F-IV. To further improve the DSC and IoU of the femur segmentation model (88.25% and 80.85% for DSC and IoU), subsequent research would experimentally modify the block of series of operation layers of the contraction and/or expansion path in the U-Net model. The limitation of this research is the tedious preparation process (i.e., time-consuming and labor-intensive nature) of the ground-truth images of the left femur necessary to train, validate, and test the U-Net left-femur model.

## Figures and Tables

**Figure 1 sensors-23-05720-f001:**
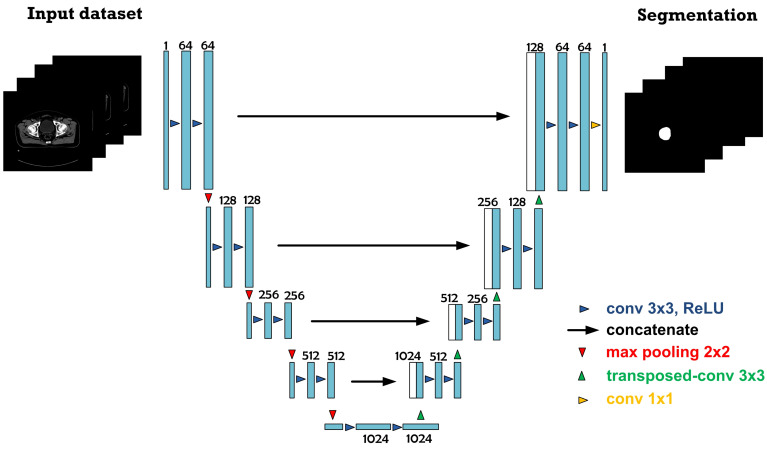
The 5-layer U-Net architecture of the deep-learning-based left-femur segmentation scheme.

**Figure 2 sensors-23-05720-f002:**
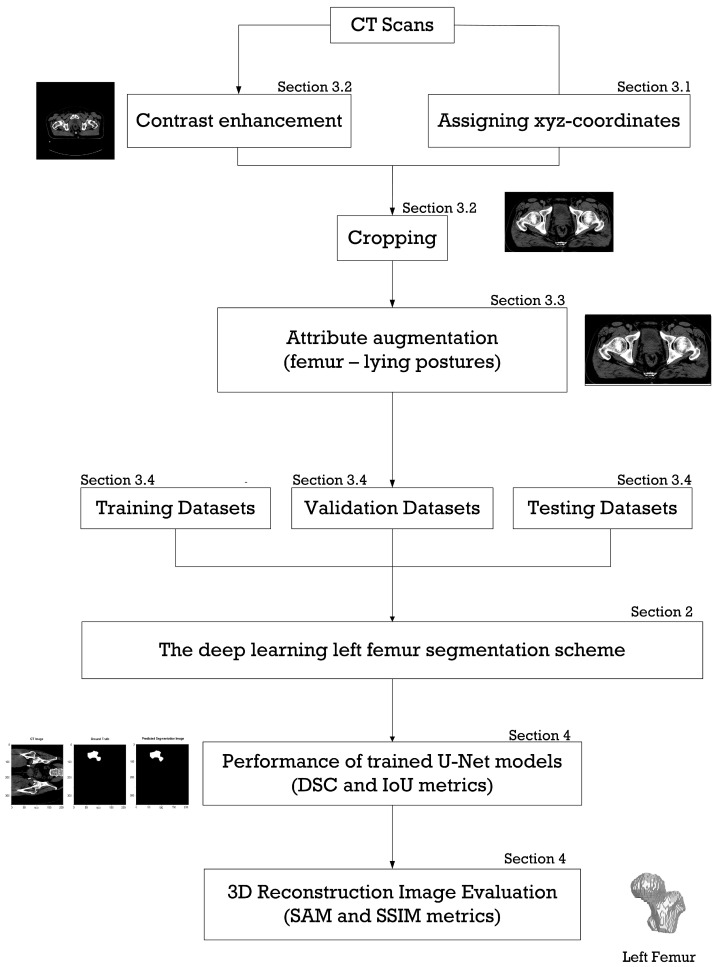
Overview of the scope of this research.

**Figure 3 sensors-23-05720-f003:**
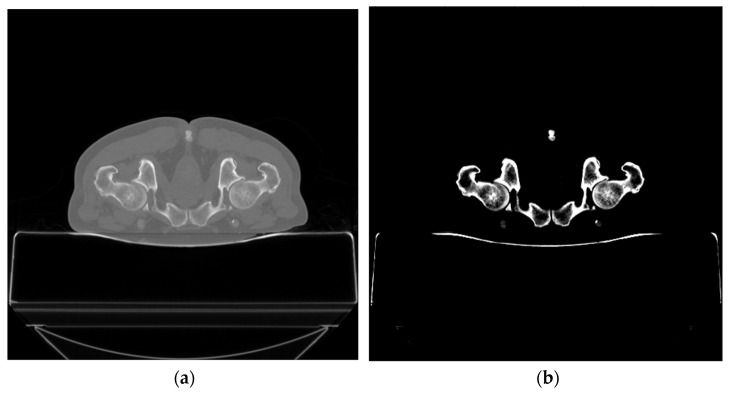
The CT slice in axial plane: (**a**) original CT slice and (**b**) CT image after window leveling and normalization (in 8-bit PNG format) with 300 HU for WL and 400 HU for WW.

**Figure 4 sensors-23-05720-f004:**
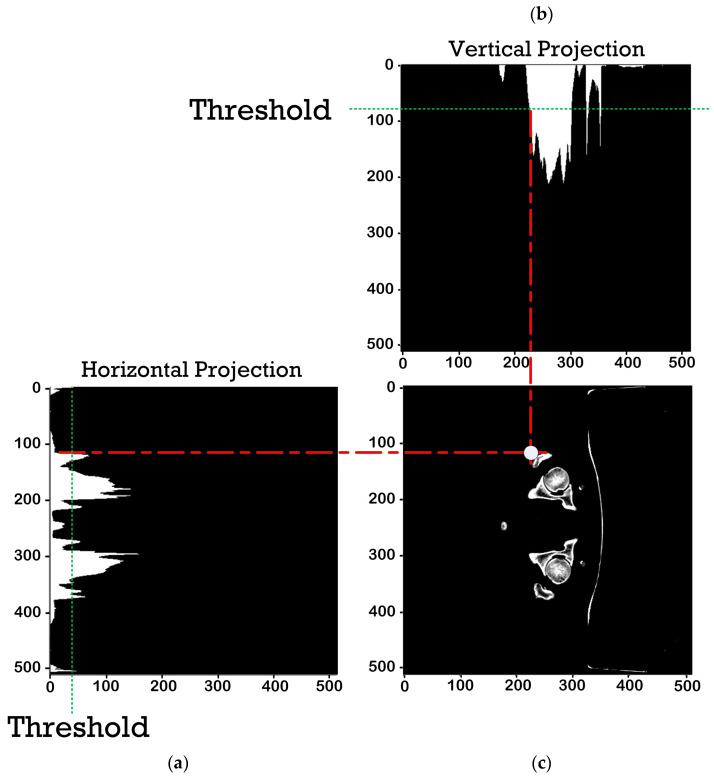
Histogram projection and xy coordinates of the femur in the axial plane: (**a**) horizontal histogram projection, (**b**) vertical histogram projection, and (**c**) xy coordinates.

**Figure 5 sensors-23-05720-f005:**
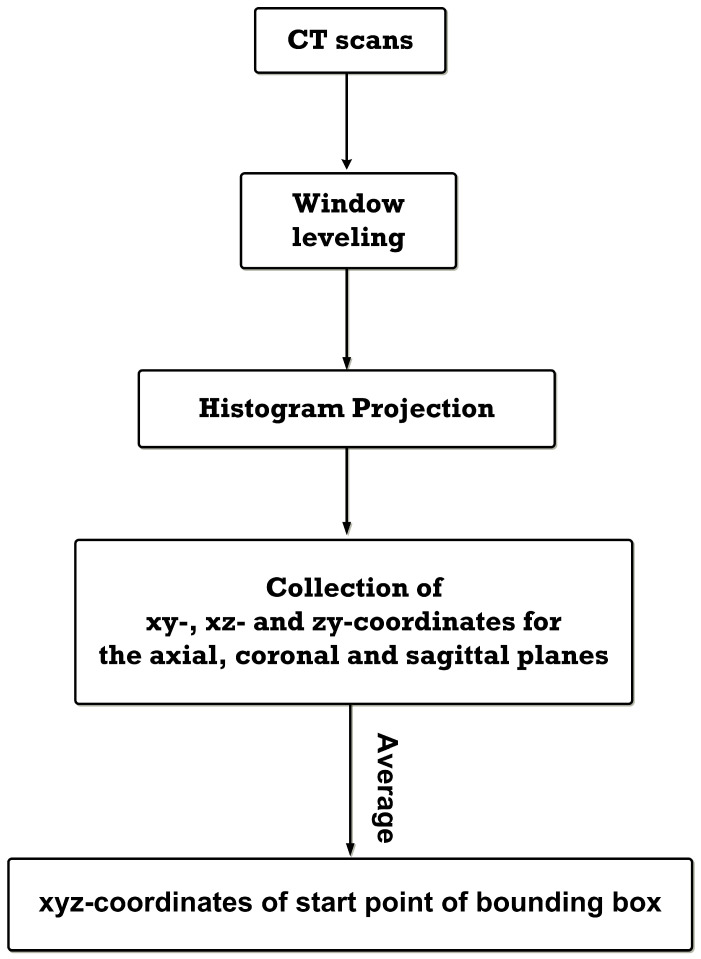
The workflow of assigning xyz coordinates of the bounding box for femur cropping.

**Figure 6 sensors-23-05720-f006:**
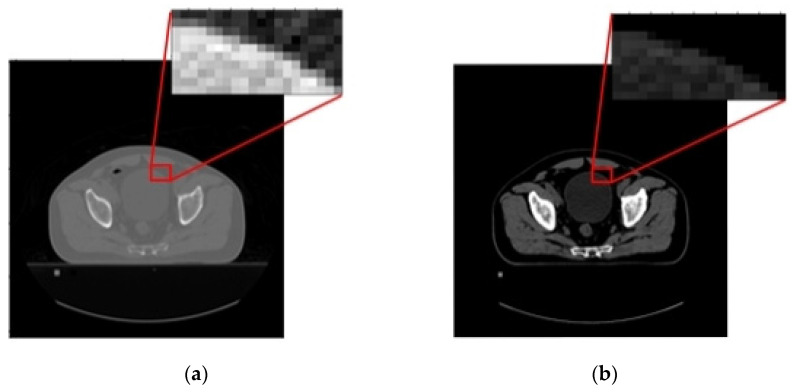
The CT slice in axial plane: (**a**) original CT slice and (**b**) CT image after window leveling and normalization (in 8-bit PNG format).

**Figure 7 sensors-23-05720-f007:**
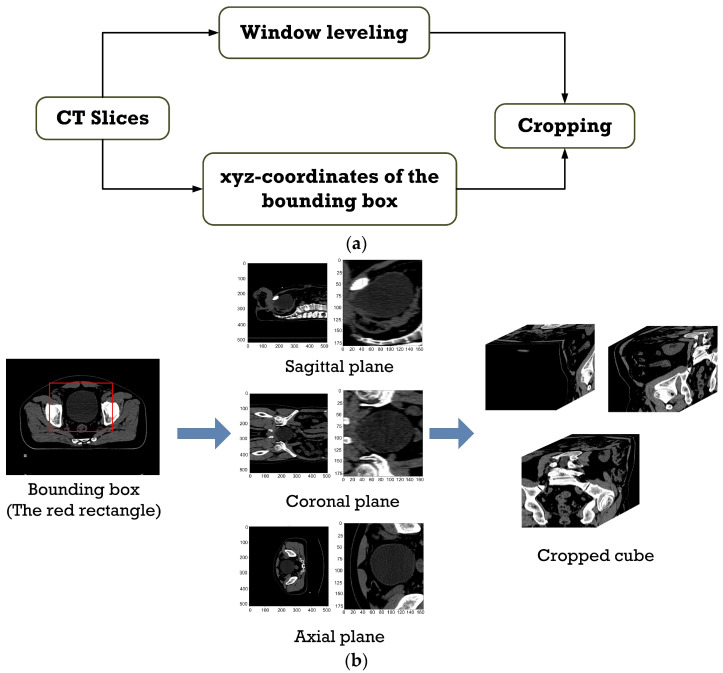
(**a**) The workflow of contrast enhancement and cropping the femur region and (**b**) the example of the bounding box for cropping and the cropped cube comprising the cropped CT slices in the axial, coronal, and sagittal planes.

**Figure 8 sensors-23-05720-f008:**
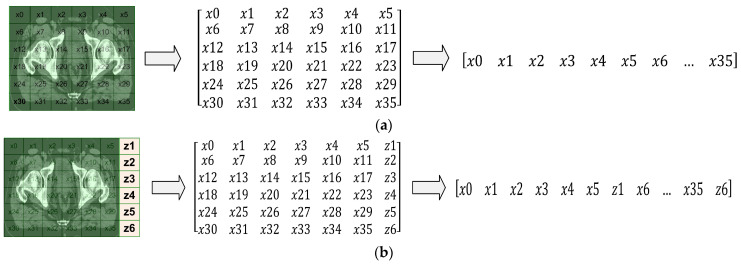
The CT image of the femur in the axial plane: (**a**) cropped CT slice and (**b**) cropped CT slice with attribute augmentation.

**Figure 9 sensors-23-05720-f009:**
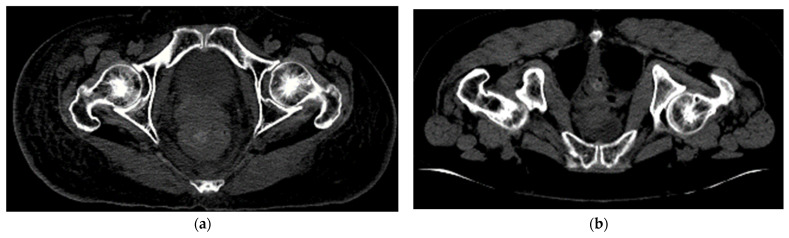
CT slices of the femur in different lying postures: (**a**) supine and (**b**) prone.

**Figure 10 sensors-23-05720-f010:**
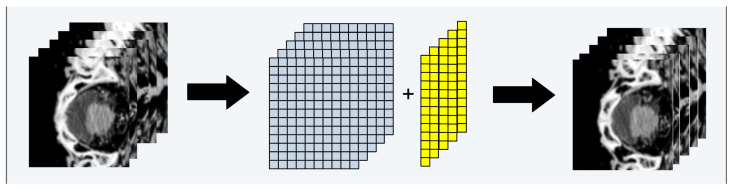
Example of attribute augmentation to the CT slices.

**Figure 11 sensors-23-05720-f011:**
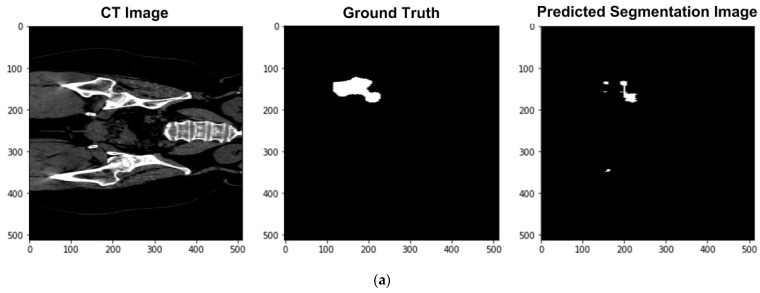
Performance of the U-Net left femur segmentation model under dataset category: (**a**) F-I, (**b**) F-II, (**c**) F-III, (**d**) F-IV, (**e**) F-V, (**f**) F-VI, (**g**) F-VII, (**h**) F-VIII.

**Figure 12 sensors-23-05720-f012:**
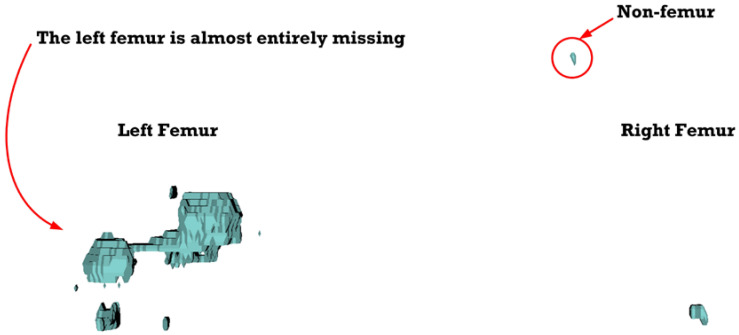
Three-dimensional reconstruction images of the proposed deep-learning-based automatic left-femur segmentation scheme under dataset category F-I.

**Figure 13 sensors-23-05720-f013:**
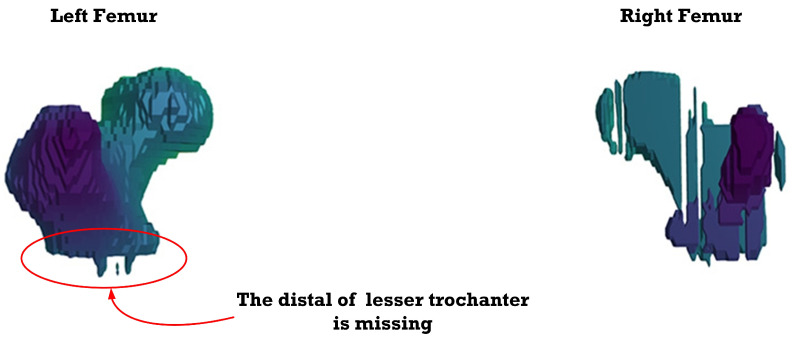
Three-dimensional reconstruction images of the proposed deep-learning-based automatic left-femur segmentation scheme under dataset category F-II.

**Figure 14 sensors-23-05720-f014:**
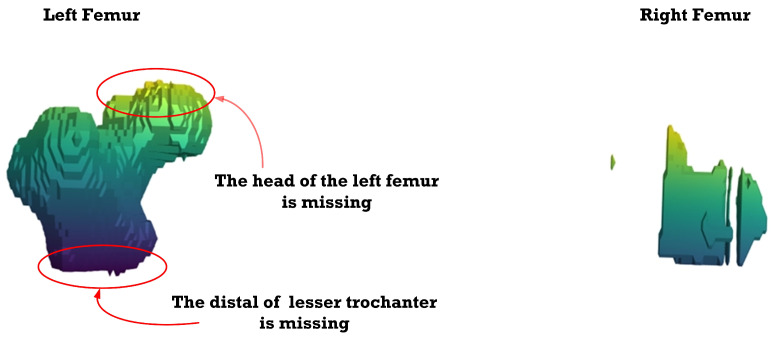
3D reconstruction images of the proposed deep-learning automatic left femur segmentation scheme under dataset category F-III.

**Figure 15 sensors-23-05720-f015:**
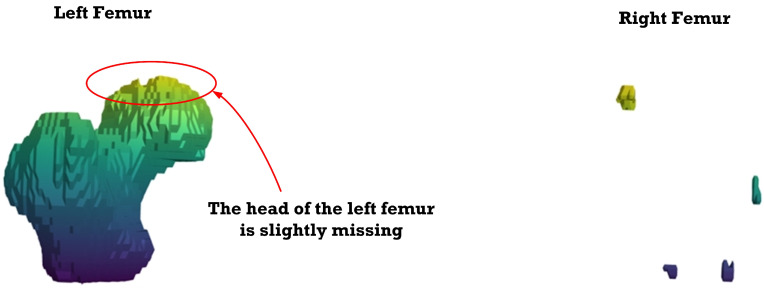
Three-dimensional reconstruction images of the proposed deep-learning-based automatic left-femur segmentation scheme under dataset category F-IV.

**Figure 16 sensors-23-05720-f016:**
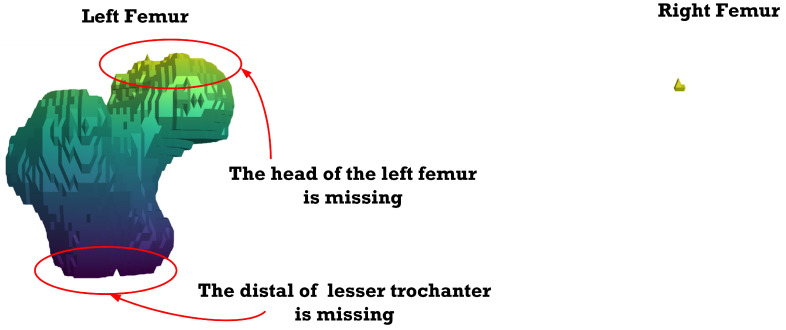
Three-dimensional reconstruction images of the proposed deep-learning-based automatic left-femur segmentation scheme under dataset category F-V.

**Figure 17 sensors-23-05720-f017:**
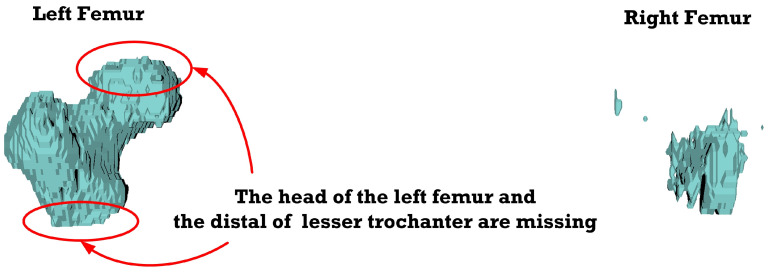
Three-dimensional reconstruction images of the proposed deep-learning-based automatic left-femur segmentation scheme under dataset category F-VI.

**Figure 18 sensors-23-05720-f018:**
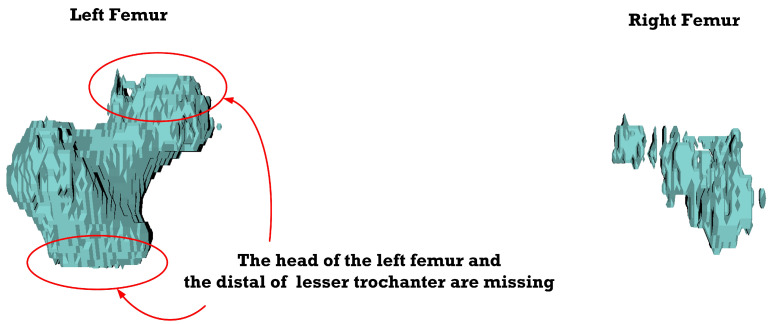
Three-dimensional reconstruction images of the proposed deep-learning-based automatic left-femur segmentation scheme under dataset category F-VII.

**Figure 19 sensors-23-05720-f019:**
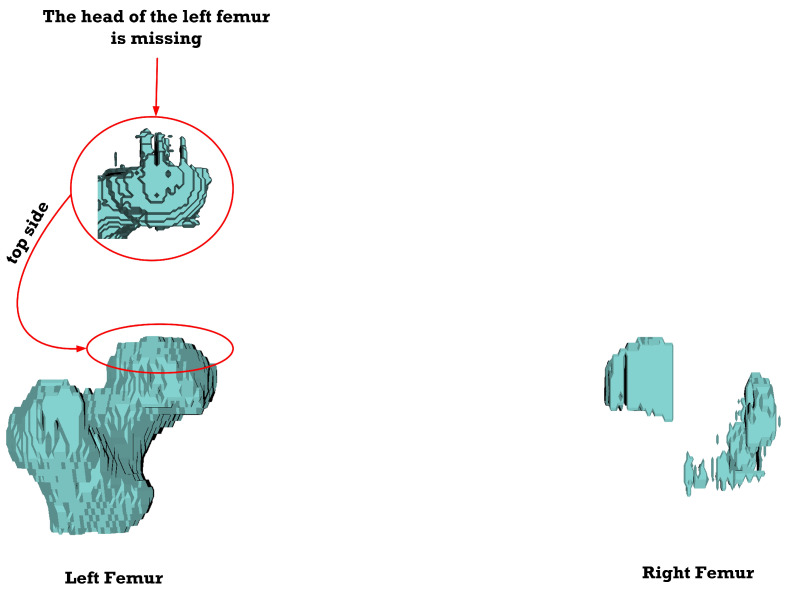
Three-dimensional reconstruction images of the proposed deep-learning-based automatic left-femur segmentation scheme under dataset category F-VIII.

**Table 1 sensors-23-05720-t001:** The initial experimental datasets of patients with lower abdominal diseases.

Organ of Interest	Left Femur	
Number of patients	120	
Age	60–80 years old	
Gender	60 male and 60 female patients	
Types of lower abdominal disorders	Cervical cancer and prostate cancer	Colorectal cancer, rectosigmoid cancer, and rectum cancer
Source of data	Siriraj Hospital, Thailand	

**Table 2 sensors-23-05720-t002:** Hyperparameters of the U-Net model for left-femur segmentation.

Hyperparameter	Value
	Femur
Number of layers	5
Epochs	5000
Learning rate	0.001
Optimizer	Adam
Loss function	Binary cross-entropy
Input dimension (pixels)	352 × 208
Convolution kernel size	3 × 3
Max pooling kernel size	2 × 2
Activation function	Rectified linear unit (ReLU), sigmoid
Initial channels	64

**Table 3 sensors-23-05720-t003:** Performance of the U-Net left-femur segmentation model in terms of DSC and IoU.

Performance of the U-Net Left-Femur Segmentation Model (%)		
Dataset Category	Metric	
	DSC	IoU
Dataset category F-I	37.76	23.90
Dataset category F-II	67.96	52.32
Dataset category F-III	61.37	45.46
Dataset category F-IV	**88.25**	**80.85**
Dataset category F-V	72.54	57.93
Dataset category F-VI	51.37	41.25
Dataset category F-VII	48.62	35.18
Dataset category F-VIII	45.27	31.30

Note: Dataset category F-I refers to the uncropped and non-augmented CT-image input datasets of the left femur; F-II to the cropped CT-image input datasets of the left femur (without attribute augmentation); F-III, F-IV, and F-V to the cropped and augmented CT-image input datasets of the left femur with small (1 and 2 for supine and prone posture), large (5 and 10), and excessively large feature coefficients (10 and 20); and F-VI, F-VII, and F-VIII to the uncropped and augmented CT-image input datasets of the left femur with small (1 and 2 for supine and prone posture), large (5 and 10), and excessively large feature coefficients (10 and 20).

**Table 4 sensors-23-05720-t004:** Image-similarity metrics (SAM and SSIM) of 3D reconstruction images of the left femur.

Isometric View of the Left Femur	Ground Truth and Predicted Results of the Left Femur under Dataset Category F-I					
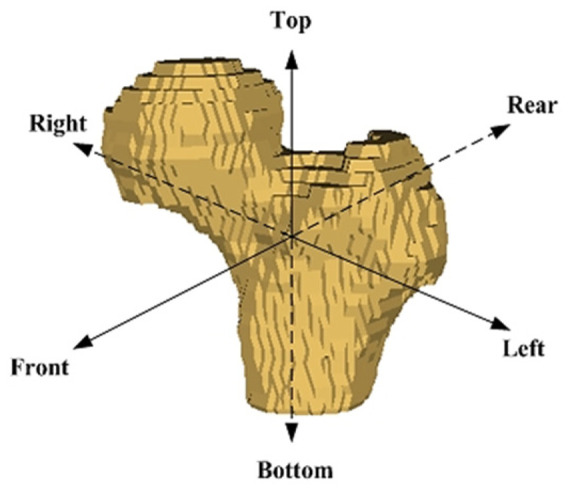	Front	Left	Rear	Right	Top	Bottom
Ground Truth	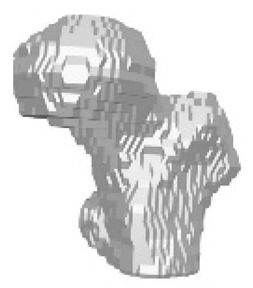	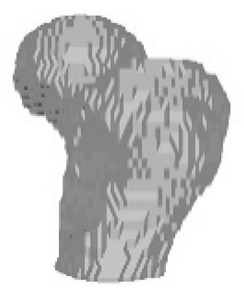	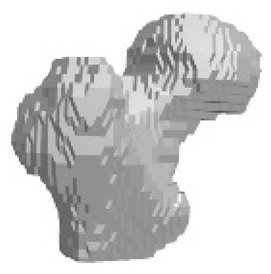	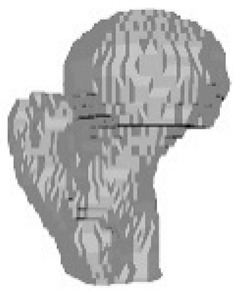	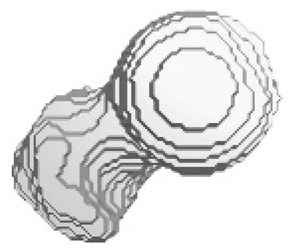	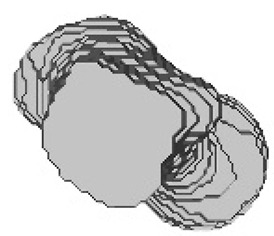
Prediction	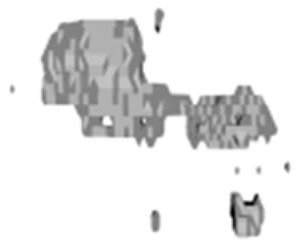	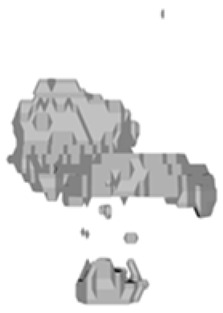	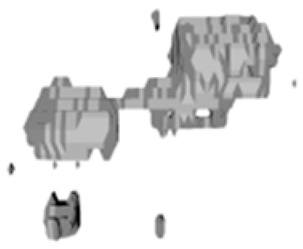	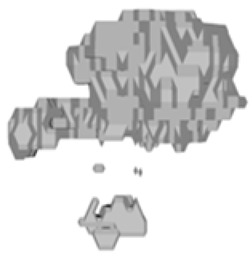	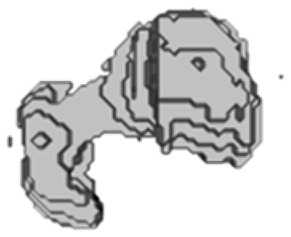	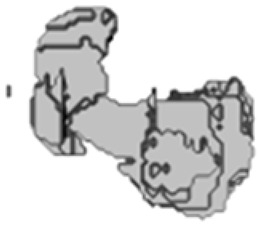
SAM	0.211	0.214	0.209	0.223	0.243	0.257
SSIM	0.433	0.454	0.501	0.532	0.502	0.544
Ground truth and predicted results of the left femur under dataset category F-II						
	Front	Left	Rear	Right	Top	Bottom
Prediction	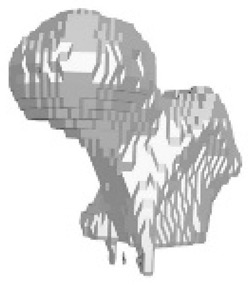	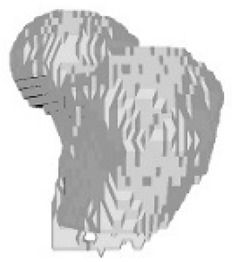	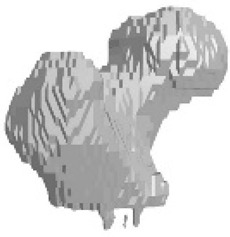	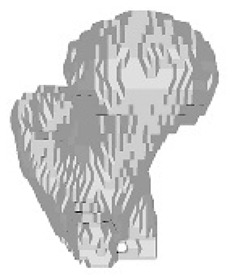	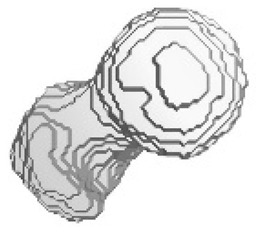	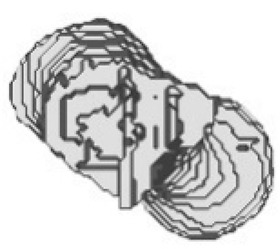
SAM	0.155	0.165	0.144	0.135	0.187	0.236
SSIM	0.677	0.658	0.712	0.714	0.701	0.689
Ground truth and predicted results of the left femur under dataset category F-III						
	Front	Left	Rear	Right	Top	Bottom
Prediction	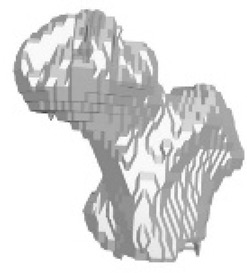	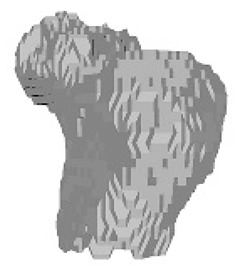	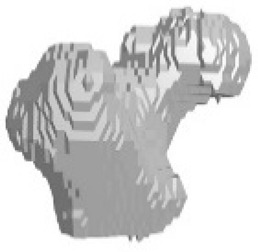	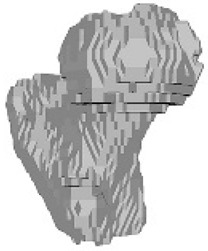	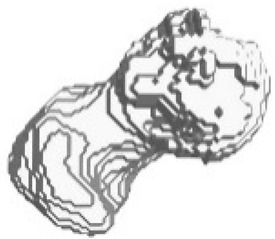	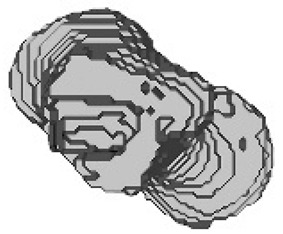
SAM	0.162	0.139	0.152	0.120	0.223	0.220
SSIM	0.668	0.669	0.708	0.729	0.688	0.708
Ground truth and predicted results of the left femur under dataset category F-IV						
	Front	Left	Rear	Right	Top	Bottom
Prediction	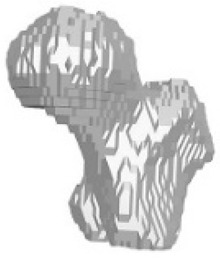	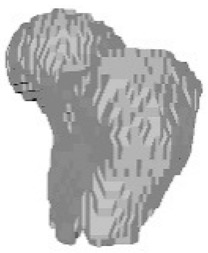	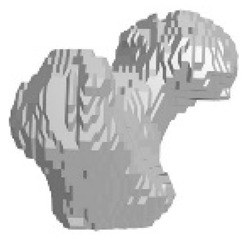	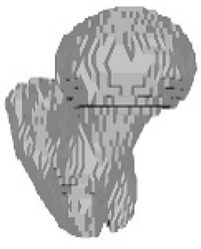	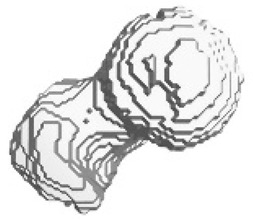	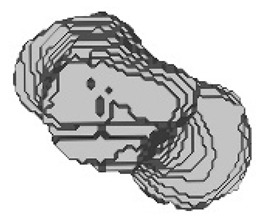
SAM	0.142	0.138	0.129	0.117	0.204	0.215
SSIM	0.702	0.706	0.725	0.732	0.701	0.710
Ground truth and predicted results of the left femur under dataset category F-V						
	Front	Left	Rear	Right	Top	Bottom
Prediction	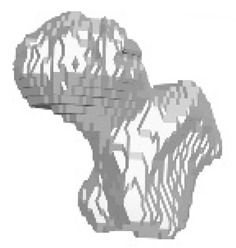	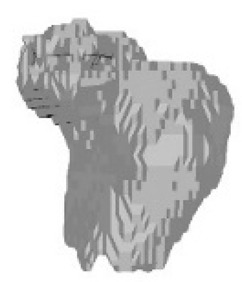	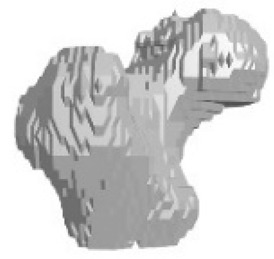	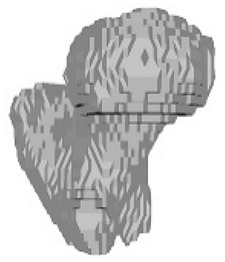	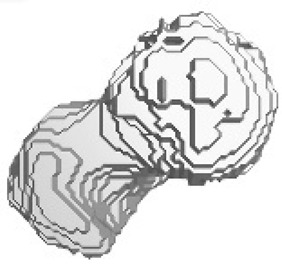	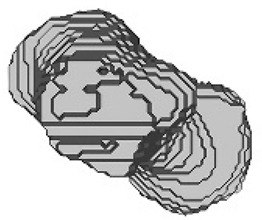
SAM	0.151	0.145	0.134	0.121	0.196	0.223
SSIM	0.683	0.675	0.722	0.729	0.700	0.702
Ground truth and predicted results of the left femur under dataset category F-VI						
	Front	Left	Rear	Right	Top	Bottom
Prediction	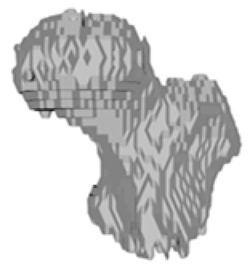	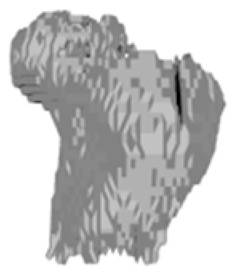	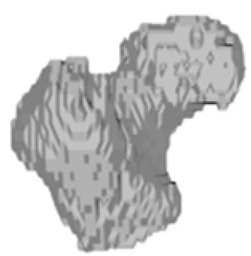	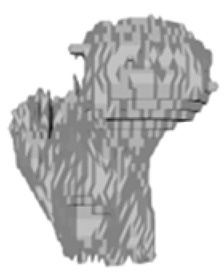	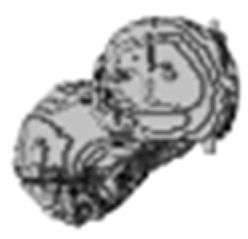	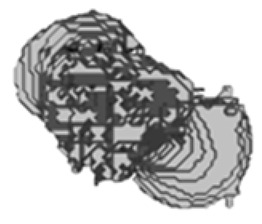
SAM	0.176	0.170	0.164	0.146	0.197	0.225
SSIM	0.596	0.572	0.632	0.634	0.644	0.635
Ground truth and predicted results of the left femur under dataset category F-VII						
	Front	Left	Rear	Right	Top	Bottom
Prediction	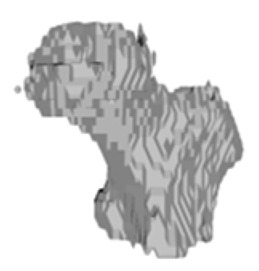	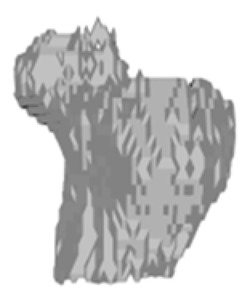	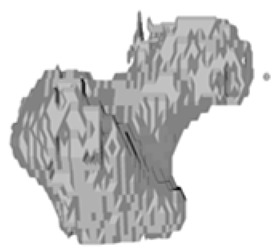	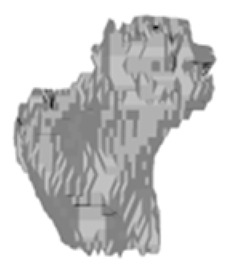	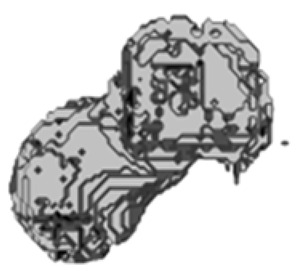	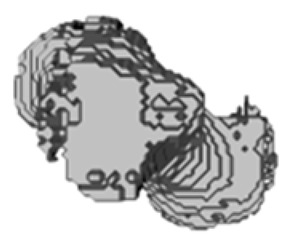
SAM	0.183	0.157	0.170	0.176	0.197	0.225
SSIM	0.551	0.572	0.631	0.603	0.637	0.633
Ground truth and predicted results of the left femur under dataset category F-VIII						
	Front	Left	Rear	Right	Top	Bottom
Prediction	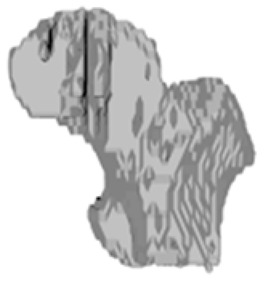	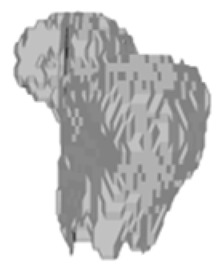	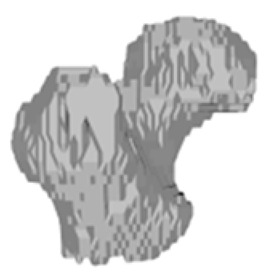	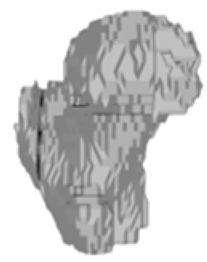	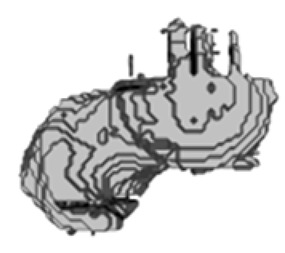	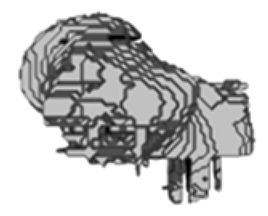
SAM	0.201	0.193	0.184	0.192	0.212	0.236
SSIM	0.461	0.471	0.531	0.574	0.536	0.639

## Data Availability

Not applicable.

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
