# Peer review of "Enhanced Deep-Learning-Based Automatic Left-Femur Segmentation Scheme with Attribute Augmentation"

_sensors, 2023, doi:10.3390/s23125720_

Round 1

Reviewer 1 Report

The authors propose augmenting cropped computed tomography (CT) slices with data attributes to enhance the performance of a deep-learning automatic multiorgan segmentation scheme.

1. The concept of using data attributes for data augmentation is indeed interesting. However, the author spends a considerable amount of text describing the well-known U-Net network structure, which most people are already familiar with. And also spends a considerable amount of text describing how to preprocess CT data, such as cropping, window leveling, contrast enhancement. The description of how to utilize data attributes for data augmentation is insufficient.

2. The author proposes using data attributes for data augmentation, such as utilizing gender attributes for bladder CT images. The method of using data attributes for data augmentation involves simply adding an attribute vector as an additional column to the original data. I have doubts about the effectiveness of this data augmentation method and whether it can truly increase the diversity of the data.

3. The results displayed in Table 3 show that B-II indicates no data augmentation was performed, while B-III indicates data augmentation using gender attributes. The results show that the Dice score for B-II is 95.15, while for B-III it is 95.46. There is no significant difference between the two. This suggests that this data augmentation method has little effect.

Reviewer 2 Report

Overall Comment: This article presents a CT data augmentation method to enhance the performance of an automatic multiorgan segmentation network. The experiments in this work were well designed and conducted, and the results are analyzed in detail. The article has the following problems that need to be revised.

Comment 1: In section 1.1, the authors mentioned the deep-learning methods for medical image segmentation, but the authors only introduced CNN-based methods. As is known, transformer-based methods have achieved superior results in medical image segmentation recently, so the transformer-based (medical) image segmentation methods should also be introduced, such as:

[1] A boundary-guided transformer for measuring distance from rectal tumor to anal verge on magnetic resonance images. Patterns. 2023 Mar 27.

[2] HCTNet: A hybrid CNN-transformer network for breast ultrasound image segmentation. Computers in Biology and Medicine. 2023 Mar 1;155:106629.

[3] H2Former: An Efficient Hierarchical Hybrid Transformer for Medical Image Segmentation. IEEE Transactions on Medical Imaging. 2023 Apr 5.

Comment 2: In section 2.1, the related works of the data augmentation methods were not introduced comprehensively enough. Before introducing their proposed method, there should be a more detailed description of its motivation.

Comment 3: For some reason, the arrow in the middle of Figure 1 looks inconsistent. The authors should revise Figure 1 further. In addition, the background color of Figure 10 is too dark, making it difficult to see what is in the middle of the figure.

Comment 4: The authors should carefully check typos in the equations, such as Eq. (7).

The writing of this manuscript is poor. The author should use the English Editing service or invite a English speaker to polish this manuscript.

Round 2

Reviewer 1 Report

The manuscript has been revised addressing the reviewer's comments.